# The Effect of High-Pressure Processing on the Survival of Non-O157 Shiga Toxin-Producing *Escherichia coli* in Steak Tartare: The Good- or Best-Case Scenario?

**DOI:** 10.3390/microorganisms11020377

**Published:** 2023-02-02

**Authors:** Josef Kameník, Marta Dušková, Kateřina Dorotíková, Markéta Hušáková, František Ježek

**Affiliations:** Department of Animal Origin Food and Gastronomic Sciences, Faculty of Veterinary Hygiene and Ecology, University of Veterinary Sciences Brno, Palackého tř. 1946/1, 612 42 Brno, Czech Republic

**Keywords:** STEC, minced meat, contamination, sous vide treatment, vacuum packing, meat color

## Abstract

Samples of steak tartare were artificially contaminated with a cocktail of Shiga toxin-producing *Escherichia coli* (STEC) O91, O146, O153, and O156 to the level of 3 log and 6 log CFU/g. Immediately after vacuum packing, high-pressure processing (HPP) was performed at 400 or 600 MPa/5 min. Some of the samples not treated with HPP were cooked under conditions of 55 °C for 1, 3, or 6 h. HPP of 400 MPa/5 min resulted in a 1–2 log reduction in the STEC count. In contrast, HPP of 600 MPa/5 min led to the elimination of STEC even when inoculated to 6 log CFU/g. Nevertheless, sub-lethally damaged cells were resuscitated after enrichment, and STEC was observed in all samples regardless of the pressure used. STEC was not detected in the samples cooked in a 55 °C water bath for 6 h, even after enrichment. Unfortunately, the temperature of 55 °C negatively affected the texture of the steak tartare. Further experiments are necessary to find an optimal treatment for steak tartare to assure its food safety while preserving the character and quality of this attractive product.

## 1. Introduction

Cooking meat is considered the most effective way of eliminating vegetative pathogenic microorganisms that cause foodborne diseases [1,2]. The combination of a temperature and time of 70 °C and 2 min guarantee a reduction in the number of non-sporogenous foodborne bacteria of more than 6 log [3].

In spite of the indisputable advantages of consuming cooked meat, there are ready-to-eat products on the market that are made from raw meat and are not cooked in any way. One of these is steak tartare. Steak tartare is a ready-to-eat food whose principal constituent is raw ground beef [4]. It is usually consumed with various sauces, vegetables, and seasonings [5]. The absence of cooking, along with product handling and the addition of other ingredients, may lead to a high level of bacterial contamination. Metagenomic analysis of steak tartare sold on the retail network identified 180 bacterial species belonging to 90 genera, and at the time of purchase, the given samples displayed a total bacterial count of as much as 7 log CFU/g in certain cases [4]. In addition to meat spoilage bacteria, potentially pathogenic bacteria such as Shiga toxin-producing *Escherichia coli* (STEC), *Salmonella enterica,* and *Listeria monocytogenes* may also occur in steak tartare [5]. The prevalence of *L. monocytogenes* was 55% in vacuum-packed steak tartare on the retail network in the Czech Republic, and 17 isolates, mostly belonging to serotype 1/2a, were obtained from samples [6]. In 2012, a foodborne disease outbreak of 24 cases, most probably linked to the consumption of steak tartare contaminated with STEC O157:H7, occurred in Belgium [7]. An outbreak caused by STEC O157 following the consumption of steak tartare has also been reported in The Netherlands [8].

Cattle represent a natural reservoir of STEC [9]. These bacteria enter the external environment, from where they may contaminate the surface of animal bodies through feces [10,11]. Abattoirs play a crucial role from the viewpoint of prevention of the cross-contamination of meat [12]. Up to 90% of cattle can have STEC on the surface of their hides, but usually, the overall rate is lower. During the technological processing of carcasses in slaughterhouses, a reduction in STEC counts may occur [13]. de Assis et al. [13] reported the highest incidence of STEC on hot carcasses (8%), whereas cold carcasses and beef samples were positive in 2% and 1% of cases, respectively. Nevertheless, the presence of STEC on the surface of beef cannot be ruled out [14].

There is a risk of human infection with STEC if beef is consumed raw, as in the case of steak tartare. Illness may also easily occur due to the fact that the ingestion of just 10–100 STEC cells is enough to cause disease [15]. Steak tartare is widely sold as a vacuum-packed product on the retail network in the Czech Republic. Sporadically, the Czech supervisory authorities report positive findings of STEC in single batches of steak tartare, resulting in its withdrawal from the market. Therefore, producers are in search of ways to prevent the occurrence of STEC in such products.

There are not many possible ways of reliably preventing the presence of STEC in raw meat without changing the character of the product or contravening valid legislation. The use of steam or hot water to treat the surface of the carcass may reduce the presence of STEC, though it does not, however, prevent possible cross-contamination during subsequent handling of the meat up until the moment at which the product is packed in its final form. In this regard, the only possible solution is to treat the product in a suitable manner after packaging when there is no further threat of secondary contamination. High-pressure processing (HPP) would seem to be a possible solution in this respect.

HPP technology is an attractive proposition since it can devitalize many of the bacteria present and thereby improve the safety and shelf life of the product without significantly affecting the sensory or nutritional properties of the food in any way [16,17]. Most of the vegetative forms of bacteria in foodstuffs are inactivated by the action of pressures of 400–600 MPa for a period of a few minutes at room temperature [18]. A pressure of 600 MPa appeared to be the most effective for the reduction of STEC during the treatment of raw meatballs made from beef [19].

The aim of this study was to determine the possibilities for the elimination of STEC in samples of artificially contaminated steak tartare using HPP (400 or 600 MPa) and to compare the effect of high pressure with heat treatment in a water bath at 55 °C.

## 2. Materials and Methods

### 2.1. Steak Tartare

The product steak tartare from producer A was used in the study. This is an uncooked meat product with a 92% proportion of beef. The product contains 20.3% protein, 6.9% fat, and 1.5% salt (mandatory nutrition declaration according to Regulation (EU) No 1169/2011). The additives used are sodium nitrite (E250), sodium acetate (E262), ascorbic acid (E300), and the thickener guar gum (E412). The protective culture SafePro^®^ B-LC-2 containing *Pediococcus acidilactici* is applied to extend shelf life and suppress *L. monocytogenes* (Chr. Hansen, Starovice, Czech Republic). Steak tartare from producer A is the most widespread product of this type on the Czech market and is offered in retail stores in vacuum skin packaging. For the purposes of the experiments conducted in this study, steak tartare was supplied directly by the producer, vacuum-packed in a quantity of several kilograms at a temperature of 1 ± 1 °C.

### 2.2. Strains of STEC Used for the Artificial Contamination of Steak Tartare

Samples of steak tartare were inoculated with a cocktail of the strains of *E. coli* O91, O146, O153, and O156 from the collection of the Department of Animal Origin Food and Gastronomic Sciences at VETUNI. These strains were stored frozen at a temperature of −80 ± 2 °C and revived on blood agar (Oxoid, Hampshire, UK) at 37 °C for 24 h. The presence of virulence factors in *E. coli* O91 (*stx*_1_, *stx*_2_), *E. coli* O146 (*stx*_1_), *E. coli* O153 (*stx*_1_, *eae*A), and *E. coli* O156 (*stx*_1_, *hly*) was verified using the multiplex PCR method [20].

### 2.3. Artificial Contamination of Steak Tartare with STEC

Two inocula of a STEC cocktail of 5 log and 8 log CFU/mL were prepared for the artificial contamination of steak tartare samples to obtain 3 log and 6 log CFU/g of STEC. McFarland turbidity standard 3 was first prepared separately in physiological solution for each strain of STEC. These suspensions were mixed, creating 8 log CFU/mL, and further diluted to an inoculum of a concentration of 5 log CFU/mL. The inoculum of 5 log CFU/mL was applied at an amount of 40 mL to 4.2 kg of steak tartare to obtain a level of 3 CFU/g, and an inoculum of 8 log CFU was applied in the same way to obtain the level of 6 CFU/g. The actual level of STEC in the inocula and in the samples of steak tartare following the application of the STEC cocktail was confirmed by cultivation at 37 °C for 24 h on Tryptone Bile X-glucuronide (TBX) agar (Oxoid).

### 2.4. Treatment of Contaminated Steak Tartare with HPP and in a Water Bath at 55 °C

The experiment was prepared as a single-batch study. Inoculation of steak tartare with STEC was followed immediately by the vacuum-packing of individual portions (100–125 g) in Cryovac^®^ CN 300 bags of a thickness of 60 μm and an OTR (oxygen transmission rate) of 13 cm^3^/m^2^/24 h/bar at 23 °C and 0% atmospheric humidity (Sealed Air Polska Sp. Zo.o., Oźarów Mazowiecki, Poland). Packing was performed with a Henkelman Lynx 32 device (Henkelman Vacuum Systems, ’s-Hertogenbosch, The Netherlands).

Immediately after packing, the samples were taken at a temperature of 3 ± 1 °C for high-pressure processing. HPP was performed on an Uhde 350-60 device (Uhde High-Pressure Technologies GmbH, Quakenbrück, Germany) at 400 MPa/5 min and 600 MPa/5 min. The temperature inside the device increased from 5 to 25 °C during the treatment. After pressurization, the samples were cooled and stored at 4 °C for 48 h and 168 h before analyses.

The heat-treated (sous vide) samples were cooked in a Softcooker Y09 water bath (La Felsinea S.R.L., Piazzola sul Brenta, Italy) under conditions of 55 °C for 1 h, 55 °C for 3 h, and 55 °C for 6 h. A Testo 104-IR thermometer (Testo s.r.o., Prague, Czech Republic) was used to monitor the temperature of the water bath. The temperature of 55 °C does not cause the denaturation of myoglobin; however, depending on the exposure time, it is sufficient to eliminate vegetative bacteria [21].

### 2.5. Microbial Enumeration

The total viable mesophilic count (TVC) and the *E. coli* count, including a qualitative determination (STEC), were determined in samples of steak tartare. A total of 225 mL of buffered peptone water (BPW; Oxoid) was added to 25 g of sample and further decimal dilutions were prepared as necessary after homogenization (Stomacher Star-BlenderTM LB 400; VWR, Leuven, Belgium). A Plate Count Agar (PCA; Oxoid) was used for the determination of the TVC at an incubation temperature of 30 °C for 72 h according to ISO 4833-1:2013 [22]. The *E. coli* count was performed on TBX agar at 37 °C for 24 h. Qualitative determination of *E. coli* was performed with inoculation from the homogenate (sample + BPW) after 24-h cultivation at 37 °C on TBX agar (37 °C for 24 h). The presence of virulence genes was tested on at least 5 colonies from each Petri dish using the multiplex PCR method [20]. Five samples of steak tartare of a weight of 100–125 g were analyzed for each inoculum and individual HPP and storage mode. Two samples of steak tartare of a weight of 100–125 g were analyzed for each inoculum in the case of heat treatment in a water bath at 55 °C.

### 2.6. CIELab Color Measurement

Color was measured using the CIE *L***a***b** system using a Konica Minolta CM-5 spectrophotometer (Konica Minolta, Japan). A measuring area of 8 mm, illuminant D65, and 10° standard observer were used. The instrument was standardized to white and black before measurement. CIE *L**—lightness, *a**—redness, *b**—yellowness were measured. Five partial measurements were measured for each sample.

### 2.7. Statistical Analysis

All data were entered into spreadsheets (Microsoft Office Excel 2019). The obtained experimental data (CFU/g) were log_10_ transformed, and the mean values and standard deviations were calculated. The differences were compared using a *t*-test because Shapiro–Wilks tests were not able to reject the normality of the data. Statistical significance was accepted at *p* < 0.05.

## 3. Results

### 3.1. Microbiological Analysis of Samples of Steak Tartare

The results of the microbiological examination of steak tartare samples after HPP treatment are shown in Table 1 and Table 2. Before inoculation, the TVC of steak tartare samples was 7.08 ± 0.02 log CFU/g on average (Table 1). This high value is a consequence of the addition of a protective culture into the product at the processing stage in the plant. In non-inoculated steak tartare samples, *E. coli* (<1.7 log CFU/g) was not detected. Nevertheless, *E. coli* bacteria were isolated following enrichment, though no virulence genes were detected by PCR and the isolates obtained did not belong to the STEC pathotype. The resultant concentrations of STEC in the inocula for the inoculation of the steak tartare were 5.30 log CFU/mL and 8.36 log CFU/mL.

It is clear from Table 2 that the action of the hydrostatic pressure of 400 MPa/5 min resulted in a reduction in the STEC count of 1–2 log, i.e., of 90–99% for both inocula and storage regimes used. Therefore, HPP of 400 Mpa/5 min was not sufficient to completely inactivate STEC in steak tartare. In contrast, the action of a pressure of 600 MPa/5 min led to the elimination of STEC even when inoculated to a level of 6 log CFU/g. Regardless of the inoculum used, there was no significant difference in STEC counts between the steak tartare samples stored for 2 days after HPP and 7 days after HPP treatment (*p* = 0.367).

Nevertheless, sub-lethally damaged cells were resuscitated following enrichment of the sample in a liquid nutrient medium, and STEC was detected in all samples artificially contaminated with STEC regardless of the pressure used during HPP.

As is clear from Table 1, HPP affected the total viable count by a maximum of 1.5 log CFU/g. The level of the STEC inoculum had practically no influence on the TVC. The TVC was approximately 1 log CFU/g lower after the action of HPP at 600 MPa than after high-pressure processing at 400 MPa.

The results of the analysis of artificially contaminated samples of steak tartare following sous vide cooking are given in Table 3.

### 3.2. The Effect of Treatment of Steak Tartare on Color Parameters

The action of a hydrostatic pressure led to changes in the color of steak tartare analyzed with the instrumental CIELab method (Table 4). Statistically significant differences (*p* < 0.01) were found between the action of pressures of 400 and 600 MPa on the parameters *L**, *a**, *b**. The value of lightness *L** and the value of *a** both increased over the parameters seen in untreated steak tartare following the action of a pressure of 400 and 600 MPa. The value of *b** in samples treated with a pressure of 400 MPa did not differ from the value recorded in untreated samples, though samples after treatment with a pressure of 600 MPa showed a lower *b** value than the untreated samples.

After cooking in a water bath at 55 °C, the parameters of CIELab deviated in the same way as after HPP treatment, i.e., the value of *L** and *a** increased, and the value of *b** decreased. The CIELab values in samples treated with HPP and in a water bath did not differ from each other (Table 4).

## 4. Discussion

### 4.1. The Effect of HPP on the Survival of STEC

According to Gareis et al. [23], there are three scenarios for the mutual relationship between bacterial foodborne agents and processed meats. If suitable barriers have been put in place, bacteria do not grow in the product and will not survive. Quantitative determination of the bacterial agent is not possible—in a “good-case scenario”, bacterial cells can be detected at the end of the investigation only following prior enrichment, and in the “best-case scenario”, the agent cannot be found even after multiplication. The next possibility is the “bad-case scenario”. The cells of foodborne bacteria cannot grow, but they survive in the product and the count of bacteria remains at its original level. In the “worst-case scenario”, the foodborne agents may grow in the given environment. An increase of several log over the initial cell level occurs [23].

In the present study, the action of a pressure of 400 MPa resulted in a reduction in STEC of approximately 1.0–1.5 log CFU/g, and the action of a pressure of 600 MPa in a reduction of 6.0 log CFU/g. When HPP of 600 Mpa/5 min was applied, it was not possible to detect STEC by direct cultivation, though the presence of STEC was proved after enrichment of the samples in a liquid medium. This was a “good-case scenario”. The results of the study are in agreement with those of the authors Black et al. [24], who artificially contaminated minced beef (80% lean content) with a cocktail of strains of STEC O157:H7 to a level of 6 log CFU/g. Samples were subjected to HPP with the use of pressures of 300, 400, and 500 MPa for a period of 10 min at temperatures of −5 and 20 °C. The treatment of minced meat with 400 MPa at −5 °C was associated with a decrease in the STEC count of 1 log CFU/g. When the same pressure was applied at 20 °C, a reduction of as much as 3 log CFU/g was observed. The STEC counts in samples stored at 4 °C remained stable for a period of 5 days [24]. In the present study, the STEC counts also did not change further in the HPP-treated products after 5-day storage at 4 °C. In ground beef patties artificially contaminated with individual strains of STEC at a concentration of 6.5 log CFU/g and treated at 400 MPa in 4 consecutive 60 s cycles, the STEC population was reduced by 2.26–4.31 log CFU/g [25].

Bernié et al. [19] managed to reduce the count of various STEC serovars in raw minced beef by 5 log CFU/g with the use of HPP at 600 MPa for 5 min. No statistically significant difference was recorded between the strains used from the viewpoint of their sensitivity. A cocktail of 4 strains of STEC was used in our study for the artificial contamination of steak tartare, of which 2 (O91 and O146) belonged among the 6 serogroups that are associated with the largest number of cases of illness in the EU [26]. Only a single strain should never be tested during experiments focusing on studying the effect of HPP on STEC [27] in view of differences in the sensitivity of strains of STEC to HPP.

Sheen et al. [17] subjected minced beef to ionization radiation (2 kGy) to eliminate contaminating *E. coli*. Samples of meat were then artificially contaminated with 39 various strains of STEC to a final level of 8 log CFU/g. This was followed by HPP at 350 MPa (4 °C) for a period of as long as 40 min. D_10_ values (conditions under which a reduction to the bacterial population of 1 log occurred) were determined with the use of Petrifilm^TM^ for each strain of STEC used. The average D_10_ value was 9.74 min, with a range from 0.89 to 25.70 min. The results obtained testify to the variability in the resistance to HPP of the strains of STEC used. The presence or absence of virulence genes (*stx*_1_, *stx*_2_, *eae*, *ehx*A) had no influence on the D_10_ value [17]. Porto-Fett et al. [28] used a cocktail of 8 STEC strains to inoculate meatballs to a level of 7 log CFU/g, which they subsequently treated with 400 or 600 MPa pressure for various lengths of time. At a pressure of 400 MPa, STEC was reduced by 0.9–1.9 log CFU/g during a 3 to 12 min application; a pressure of 600 Mpa, when applied for a duration of 0.5 to 3 min, reduced the level of STEC by 1.4–2.9 log CFU/g. In beef meatballs, the D_10_ value was 6.54 min at a pressure of 400 MPa, and only 1.45 min at a pressure of 600 MPa [28].

The authors Hsu et al. [29] compared the ability of HPP to inactivate STEC belonging to the “Big Six” non-O157:H7 as compared to the serovar O157:H7 in minced beef. Samples were treated with a pressure of 250, 350, and 450 MPa lasting for 5, 15, or 30 min at 4 °C. The strain of serovar O157:H7 was more resistant to HPP than the non-O157:H7 strains. The overall variability in the resistance of bacteria to stress conditions may be caused by changes in the expression or activity of the sigma factor. This is the protein coded by the gene *rpo*S [30]. It can regulate the expression of around 10% of *E. coli* genes, the majority of which contribute to the general response of the cells to stress, including resistance to HPP. A large proportion of isolates of STEC are resistant to pressures as high as 600 MPa [31]. *rpoS* is a gene that is highly liable to mutations in the *E. coli* population [30]. The studies by Gayán et al. [32] revealed a set of genes and operons (*hdf*R, *rbs*, and *suc*) whose importance to the survival of HPP was previously unknown. By deleting *suc*C/D, the authors managed to significantly improve the resistance of *E. coli* to HPP by means of a mechanism based on the increased activity of RpoS. They concluded from the results obtained that mechanisms of resistance to HPP and resistance to heat are not necessarily functionally equivalent [32].

This study confirmed the conclusions reached by Diez et al. [33], indicating that HPP technology is effective at suppressing Gram-negative bacteria, while Gram-positive microorganisms are more resistant to higher pressures. The TVC was only 1.5–2.0 log CFU/g lower after treatment at 600 MPa. In contrast, the reduction in STEC was 6.0 log CFU/g. Vercammen et al. [34] tested the use of HPP on cooked hams made from pork. Various values of pressure (100–700 MPa/10 min) were used on bacteria causing spoilage and on selected foodborne agents spread on slices of ham at a level of around 8 log CFU/g. The tested bacteria were, in the majority of cases, beneath the limit of detection following the application of a pressure of 500 MPa and higher. Exceptions to this were *Latilactobacillus sakei* (2.1 log CFU/g after treatment with 600 MPa/5 °C) and *Brochothrix thermosphacta* (2.1 log CFU/g after 700 MPa/5 °C).

### 4.2. The Survival of STEC in a Water Bath at 55 °C

The lethal effects of heat treatment begin at a temperature of around 55 °C [21]. Barbosa et al. [35] found a reduction of *E. coli* in artificially inoculated hamburgers (6.59 ± 0.15 CFU/g) after heat treatment to medium-rare with an internal temperature of 57.0 ± 2.0 °C by approximately 2 log CFU/g. Ferigolo et al. [36] assessed the survival of *E. coli* in beef sirloin medallions (100 ± 5 g) after artificial contamination at 6–7 log CFU/g using the sous vide method at 54 °C. No cells of *E. coli* were detectable after 9 h. In our study, the STEC counts in heat-treated samples were under the limit of detection (<1.70 CFU/g) for both concentrations (3 and 6 log CFU/g). After enrichment, STEC was detected in steak tartare heat treated for 1 h in the case of 3 log CFU/g. In the case of the concentration of 6 log CFU/g, STEC survived the action of 55 °C/1 h and 55 °C/3 h treatment (Table 3). STEC was not detected at all, even following enrichment, after 6 h of cooking. This form of treatment (55 °C/6 h) can, therefore, also be considered “the best-case scenario” from the viewpoint of food safety. However, when processing steak tartare samples after cooking at 55 °C, a change in texture due to the denaturation of muscle proteins was evident. As a result, steak tartare lost its spreadability, which is an essential feature of this type of product. Therefore, this type of treatment is not relevant for practical use.

### 4.3. The Effect of Treatment of Steak Tartare on the Color of the Product

An increase in the *L** value, i.e., a lightness, in samples of minced beef following treatment with high pressure has been described by Black et al. [24] and Zhou et al. [27] and is in agreement with the results of this study. According to Bolumar et al. [18], structural modifications lead to changes in the ratios of absorbed, diffracted, and reflected light, which results in the increased scattering of light and, thereby, the paler appearance of the meat. Meat following HPP is extremely similar in appearance to cooked meat, as is confirmed by the results given in Table 4.

The color of meat treated with nitrite/nitrate (*cured meat*) is generally said to be significantly more stable during HPP [18]. Kameník et al. [37] did not find statistically significant differences between products treated with a high pressure of 600 MPa and untreated control samples during instrumental analysis of the color of samples of cooked comminuted meat products. Bajovic et al. [38], however, mentioned numerous studies that have described an increase in lightness *L* and a fall in the value of red color *a** in meat products after high-pressure processing. Del Olmo et al. [39] did not find any statistically significant differences in color during their testing of the effect of HPP on the product “lacón”. The stability of the color of heat-treated meat products with nitrite is caused by the denaturation of nitrosomyoglobin and the formation of stable nitrosohemochrome. However, this chemical reaction did not occur in steak tartare after the addition of nitrite without heat treatment, which could be the reason for the higher value of *L** after HPP treatment.

## 5. Conclusions

Treatment of artificially contaminated samples of steak tartare with a high pressure of 400 or 600 MPa was successful in reducing the STEC count, with a reduction of 6 log CFU/g after treatment with 600 MPa. We must, nevertheless, consider this result a “good-case scenario” since STEC could again be detected following enrichment of the sample. In contrast, STEC could not be found after the cooking of samples in a water bath at a temperature of 55 °C for 6 h even after enrichment, and this treatment can, therefore, be considered the “best-case scenario”.

Consideration must, however, also be given to the effect on the sensory properties of steak tartare. In the case of a temperature of 55 °C, this meant, first and foremost, a change in texture caused by the denaturation of proteins which had a fundamental effect on the character of the product. Further experimentation with the action of high pressure, and not merely from the perspective of the level of MPa used but also in view of the length of action and cyclicality, will be necessary for the search for the optimal treatment of steak tartare for the purpose of assuring food safety while preserving the character and quality of this attractive product.

## Figures and Tables

**Table 1 microorganisms-11-00377-t001:** Results for the TVC (mean ± standard deviation) in CFU/g in the product steak tartare following artificial inoculation with a cocktail of STEC of 3 and 6 log CFU/g and following pasteurization using HPP (high-pressure processing) at a pressure of 400 and 600 MPa.

Treatment Mode (*n* = 5)	TVC (log CFU/g)	
	STEC Inoculum 3 log CFU/g	STEC Inoculum 6 log CFU/g
before STEC inoculation	7.08 ± 0.02 ^a^	
after inoculation before HPP	7.07 ± 0.12 ^a^	7.29 ± 0.02 ^b^
400 MPa/5 min (48 h after HPP)	6.61 ± 0.04 ^c^	6.64 ± 0.03 ^c^
400 MPa/5 min (168 h after HPP)	6.55 ± 0.03 ^d^	6.66 ± 0.04 ^c^
600 MPa/5 min (48 h after HPP)	5.67 ± 0.04 ^e^	5.59 ± 0.06 ^e^
600 MPa/5 min (168 h after HPP)	5.62 ± 0.04 ^e^	5.66 ± 0.07 ^e^

^a,b,c,d,e^ log CFU/g followed by the same lower-case letter in the column did not differ significantly (*p* > 0.05).

**Table 2 microorganisms-11-00377-t002:** Results for the STEC count (mean ± standard deviation) in CFU/g in the product steak tartare following artificial inoculation with a cocktail of STEC of 3 and 6 log CFU/g and following pasteurization using HPP (high-pressure processing) at a pressure of 400 and 600 MPa.

Treatment Mode (*n* = 5)	STEC Concentration (log CFU/g)	
	STEC Inoculum 3 log CFU/g	STEC Inoculum 6 log CFU/g
after inoculation before HPP	3.80 ± 0.06 ^a^	6.39 ± 0.14 ^a^
400 MPa/5 min (48 h after HPP)	2.18 ± 0.29 ^b^	4.67 ± 0.06 ^b^
400 MPa/5 min (168 h after HPP)	1.90 ± 0.28 ^b^	4.59 ± 0.19 ^b^
600 MPa/5 min (48 h after HPP)	ND	ND
600 MPa/5 min (168 h after HPP)	ND	ND

ND: not detected (<1.70 log CFU/g); ^a,b^ log CFU/g followed by the same lower-case letter in the column did not differ significantly (*p* > 0.05).

**Table 3 microorganisms-11-00377-t003:** Results for the STEC count (mean ± standard deviation) in CFU/g in the product steak tartare following artificial inoculation with a cocktail of STEC of 3 and 6 log CFU/g and following sous vide cooking at 55 °C for a period of 1, 3, and 6 h.

Treatment Mode (*n* = 2)	STEC Concentration (log CFU/g)	
	STEC Inoculum 3 log CFU/g	STEC Inoculum 6 log CFU/g
after STEC inoculation	3.80 ± 0.06	6.39 ± 0.14
55 °C/1 h	ND	ND
55 °C/3 h	ND	ND
55 °C/6 h	ND	ND
	Determination of STEC after enrichment *	
55 °C/1 h	2/2	2/2
55 °C/3 h	0/2	2/2
55 °C/6 h	0/2	0/2

ND: not detected (<1.70 log CFU/g); * data expressed as the number of positive samples (presence in 25 g) per number of investigated samples.

**Table 4 microorganisms-11-00377-t004:** Values of *L**, *a**, *b** during instrumental analysis of the color (CIELab) of samples of the product steak tartare treated by HPP (high-pressure processing) at 400 and 600 MPa or heat-treated at 55 °C.

Parameters of Sample	CIELab Parameters					
	*L**		*a**		*b**	
**untreated sample**	**45.78**		**10.34**		**13.35**	
**HPP**	**400 MPa**	**600 MPa**	**400 MPa**	**600 MPa**	**400 MPa**	**600 MPa**
3 log/48 h	51.97	53.23	15.73	14.83	12.42	11.88
6 log/48 h	50.90	52.68	18.19	16.37	13.90	12.69
3 log/168 h	51.37	53.72	16.33	14.63	13.09	12.06
6 log/168 h	50.96	51.88	14.53	10.96	12.89	12.09
	*p* = 0.0002		*p* = 0.003		*p* = 0.003	
water bath	** *L* ** *****		** *a* ** *****		** *b* ** *****	
55 °C/1 h	51.70		14.12		11.05	
55 °C/3 h	52.20		14.69		11.06	
55 °C/6 h	51.50		14.56		10.21	

## Data Availability

The data presented in this study are available on request from the corresponding author. The data are not publicly available due to privacy.

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
