# Peer review of "The Effect of High-Pressure Processing on the Survival of Non-O157 Shiga Toxin-Producing *Escherichia coli* in Steak Tartare: The Good- or Best-Case Scenario?"

_microorganisms, 2023, doi:10.3390/microorganisms11020377_

Round 1

Reviewer 1 Report

In my opinion, this research paper contains significant information that justify publication, with major revision regarding the following issues:

1.     Abstract

In page 2 line 16 - acronym: “... of STEC (O91, ....)...” should be “...Shiga toxin-producing Escherichia coli (STEC) ...”.

Attention should be paid to the clarity of expression and readability. The paper should be revised by a native English to better understanding and adequate scientific writing, for example:

Page 2 line 23 and 24 – “ ... and STEC was demonstrated in all samples ...”, you  mean “ ... and STEC was observed in all samples...” ? , this type of writing can be seen along the paper, namely in results and discussion sections.

2.     Introduction - The content is succinctly described and contextualized with respect to previous and present theoretical background on the topic and supported by relevant references on the topic. Objectives of the study are clearly defined, although results analysis and discussion can be improved in order to clarify some aspects concerning the objective of “compare the effect of a high pressure with heat treatment in a water bath at 55 °C.”

Regarding relationship to literature, this paper has cited relevant literature in the field, although some relevant literature should be paid attention, namely:

i)                    Barbosa, A. D.; Alexandre, B.; Tondo, E.C., Malheiros, P.S. Microbial survival in gourmet hamburger thermally processed by different degrees of doneness, International Journal of Gastronomy and Food Science. 2022, 100501, https://doi.org/10.1016/j.ijgfs.2022.100501.

ii)                   Porto-Fett ACS, Jackson-Davis A, Kassama LS, Daniel M, Oliver M, Jung Y, Luchansky JB. Inactivation of Shiga Toxin-Producing Escherichia coli in Refrigerated and Frozen Meatballs Using High Pressure Processing. Microorganisms. 2020 Mar 3;8(3):360. doi: 10.3390/microorganisms8030360.

iii)                 Jiang Y, Scheinberg JA, Senevirathne R, Cutter CN. The efficacy of short and repeated high-pressure processing treatments on the reduction of non-O157:H7 Shiga-toxin producing Escherichia coli in ground beef patties. Meat Sci. 2015 Apr;102:22-6. doi: 10.1016/j.meatsci.2014.12.001.

iv)                 Wasilenko JL, Fratamico PM, Sommers C, DeMarco DR, Varkey S, Rhoden K, Tice G. Detection of Shiga toxin-producing Escherichia coli (STEC) O157:H7, O26, O45, O103, O111, O121, and O145, and Salmonella in retail raw ground beef using the DuPont™ BAX® system. Front Cell Infect Microbiol. 2014 Jun 18;4:81. doi: 10.3389/fcimb.2014.00081.  

3.     Materials and Methods: The methodology used is appropriate and adjusted to the objectives of this work and presented with detail, although some aspects must be revised:

-          Page 4 line 90-91 – the product composition presented is in mean values, the information source is missing

-          How many production batches were used in this study, in section 2.1 is missing ...”this information. This is a single batch study?

-          +age 4 line 109- 110 – information is repeated : “Two inocula of a STEC cocktail of 5 log and 8 log CFU/mL concentrations were prepared for the artificial contamination of steak tartare samples. STEC cocktail inocula of a 5 log and 8 log

-          CFU/mL were prepared for the artificial contamination of steak tartare samples to obtain 3 log and 6 log CFU/g of STEC in steak tartare samples. “

-          Page 4 line 115 – “...The inocula were applied at an amount of 40 mL to 4.2 kg of steak tartare....” – how many replica were prepared?

-          Page 5 – line 117 – “... TBX agar” – reference of trademark/ brand is missing and it should be written  “... Tryptone Bile X-glucuronide (TBX) Agar  ( trademark reference…” to be in accordance with other agar medium that are referred in the present work.

-          “ ... 400 MPa/5 min. and 600 MPa/5 min. within a temperature range of 5–25 °C...” – which temperature was used which temperatures conditions were tested

-          Regarding Treatment of contaminated steak tartare with HPP and in a water bath at 55 °C  (section 2.4) it is not clear how many samples per condition were prepared. Please clarify

Regarding the Treatment of contaminated steak tartare in a water bath at 55 °C, what was the criteria used for the definition of the treatment time? And it is not mentioned how the samples were stored before analysis.

-          Page 5 line 137 – “The total viable count (TVC) and ... “ – this is a mesophilic bacterial counts? Please clarify

-          Page 5 line 141  “ PCA reference is not complete.

-          Page 6 – line 157 “ ... Five partial measurements were measured for each sample. Differences in the results between the analysed samples were compared by a t-test (MS Excel)... “ – this statistical treatment was carried out to validate replicates, right?

-           Is missing a statistical analysis section – no information is given concerning the statistical methodology used to compare treatments, and treatment conditions within the same treatment.

4.     Results

-          Please revise the text of Tables 1, 2, 3 and 4, the legend is too extended, please revise in order to be less confused, for instance in Table 1 there is no need the information “Five samples of steak tartare of a weight of 100–125 g were analysed for each inoculum”, (it should be in the Materials and Methods)

-          174 and each individual mode of HPP and storage.

-          Attention should be paid to units – please revise all the Table correcting the time unit (h instead of h.; min instead of min.), please revise the text along the results and discussion section (e.g. line 176, 179)

Table 1 – statistical information is missing regarding the treatment conditions comparison, P values are presented but no information is given in order to identify the treatments presenting significant differences. Table 2 , Table 3 and 4, have the same problem.

Table 3 – it is not clear what is the “Determination of STEC after multiplication”, what is the importance, or reliability for the study when used reduce number of samples. Is it reliable?

Table 4 – please correct the “B”, it should be “b”

The results are weakly supported due to the lack of statistical analysis, or at least poorly presented.

5 – Discussion

In section 4.2, line 302-306 – “---when processing steak tartare samples after cooking at 55 °C, a change in texture due to denaturation of muscle proteins was evident. As a result, steak tartare lost its spreadability, which is an essential feature for this type of product. Therefore, this type of treatment is not relevant for practical use.” – as previously asked, what were the criteria used to define the treatments conditions? Pease clarify.

Author Response

Response to Reviewer #1

We would like to thank you for your comments and suggestions that have improved the quality of our manuscript.

Comments and Suggestions for Authors

 In my opinion, this research paper contains significant information that justify publication, with major revision regarding the following issues:

  1. Abstract

In page 2 line 16 - acronym: “... of STEC (O91, ....)...” should be “...Shiga toxin-producing Escherichia coli (STEC) ...”.

Attention should be paid to the clarity of expression and readability. The paper should be revised by a native English to better understanding and adequate scientific writing, for example:

Page 2 line 23 and 24 – “ ... and STEC was demonstrated in all samples ...”, you  mean “ ... and STEC was observed in all samples...” ? , this type of writing can be seen along the paper, namely in results and discussion sections.

We agree. These comments were corrected as suggested by the reviewer.

  1. Introduction - The content is succinctly described and contextualized with respect to previous and present theoretical background on the topic and supported by relevant references on the topic. Objectives of the study are clearly defined, although results analysis and discussion can be improved in order to clarify some aspects concerning the objective of “compare the effect of a high pressure with heat treatment in a water bath at 55 °C.”

Regarding relationship to literature, this paper has cited relevant literature in the field, although some relevant literature should be paid attention, namely:

  1. Barbosa, A. D.; Alexandre, B.; Tondo, E.C., Malheiros, P.S. Microbial survival in gourmet hamburger thermally processed by different degrees of doneness, International Journal of Gastronomy and Food Science. 2022, 100501, https://doi.org/10.1016/j.ijgfs.2022.100501.
  2. Porto-Fett ACS, Jackson-Davis A, Kassama LS, Daniel M, Oliver M, Jung Y, Luchansky JB. Inactivation of Shiga Toxin-Producing Escherichia coli in Refrigerated and Frozen Meatballs Using High Pressure Processing. Microorganisms. 2020 Mar 3;8(3):360. doi: 10.3390/microorganisms8030360.
  • Jiang Y, Scheinberg JA, Senevirathne R, Cutter CN. The efficacy of short and repeated high-pressure processing treatments on the reduction of non-O157:H7 Shiga-toxin producing Escherichia coli in ground beef patties. Meat Sci. 2015 Apr;102:22-6. doi: 10.1016/j.meatsci.2014.12.001.
  1. Wasilenko JL, Fratamico PM, Sommers C, DeMarco DR, Varkey S, Rhoden K, Tice G. Detection of Shiga toxin-producing Escherichia coli (STEC) O157:H7, O26, O45, O103, O111, O121, and O145, and Salmonella in retail raw ground beef using the DuPont™ BAX® system. Front Cell Infect Microbiol. 2014 Jun 18;4:81. doi: 10.3389/fcimb.2014.00081.

The text was edited, including recommended relevant literature.

  1. Materials and Methods: The methodology used is appropriate and adjusted to the objectives of this work and presented with detail, although some aspects must be revised:

- Page 4 line 90-91 – the product composition presented is in mean values, the information source is missing

- How many production batches were used in this study, in section 2.1 is missing ...”this information. This is a single batch study?

- Page 4 line 109- 110 – information is repeated : “Two inocula of a STEC cocktail of 5 log and 8 log CFU/mL concentrations were prepared for the artificial contamination of steak tartare samples. STEC cocktail inocula of a 5 log and 8 log

- CFU/mL were prepared for the artificial contamination of steak tartare samples to obtain 3 log and 6 log CFU/g of STEC in steak tartare samples. “

- Page 4 line 115 – “...The inocula were applied at an amount of 40 mL to 4.2 kg of steak tartare....” – how many replica were prepared?

- Page 5 – line 117 – “... TBX agar” – reference of trademark/ brand is missing and it should be written  “... Tryptone Bile X-glucuronide (TBX) Agar  ( trademark reference…” to be in accordance with other agar medium that are referred in the present work.

- “ ... 400 MPa/5 min. and 600 MPa/5 min. within a temperature range of 5–25 °C...” – which temperature was used which temperatures conditions were tested

- Regarding Treatment of contaminated steak tartare with HPP and in a water bath at 55 °C  (section 2.4) it is not clear how many samples per condition were prepared. Please clarify

Regarding the Treatment of contaminated steak tartare in a water bath at 55 °C, what was the criteria used for the definition of the treatment time? And it is not mentioned how the samples were stored before analysis.

- Page 5 line 137 – “The total viable count (TVC) and ... “ – this is a mesophilic bacterial counts? Please clarify

- Page 5 line 141  “ PCA reference is not complete.

- Page 6 – line 157 “ ... Five partial measurements were measured for each sample. Differences in the results between the analysed samples were compared by a t-test (MS Excel)... “ – this statistical treatment was carried out to validate replicates, right?

- Is missing a statistical analysis section – no information is given concerning the statistical methodology used to compare treatments, and treatment conditions within the same treatment.

We agree. These comments were corrected as suggested by the reviewer.

  1. Results

- Please revise the text of Tables 1, 2, 3 and 4, the legend is too extended, please revise in order to be less confused, for instance in Table 1 there is no need the information “Five samples of steak tartare of a weight of 100–125 g were analysed for each inoculum”, (it should be in the Materials and Methods)

- 174 and each individual mode of HPP and storage.

- Attention should be paid to units – please revise all the Table correcting the time unit (h instead of h.; min instead of min.), please revise the text along the results and discussion section (e.g. line 176, 179)

We agree. These comments were corrected as suggested by the reviewer.

Table 1 – statistical information is missing regarding the treatment conditions comparison, P values are presented but no information is given in order to identify the treatments presenting significant differences. Table 2 , Table 3 and 4, have the same problem.

Table 3 – it is not clear what is the “Determination of STEC after multiplication”, what is the importance, or reliability for the study when used reduce number of samples. Is it reliable?

Table 4 – please correct the “B”, it should be “b”

The results are weakly supported due to the lack of statistical analysis, or at least poorly presented.

We agree. These comments were corrected as suggested by the reviewer. Determination of STEC after multiplication (Table 3) is important to rate whether it is a Good- or Best-Case Scenario.

5 – Discussion

In section 4.2, line 302-306 – “---when processing steak tartare samples after cooking at 55 °C, a change in texture due to denaturation of muscle proteins was evident. As a result, steak tartare lost its spreadability, which is an essential feature for this type of product. Therefore, this type of treatment is not relevant for practical use.” – as previously asked, what were the criteria used to define the treatments conditions? Pease clarify.

The text was supplemented according to this comment.

Reviewer 2 Report

Following is the review of manuscript entitled “The Effect of High Pressure Processing on the Survival of Non-O157 Shiga  Toxin-producing Escherichia coli in Steak Tartare: The Good- or Best-Case  Scenario?”.  The authors have presented data on treatment of steak tartare under high pressure and in combination of heat treatment. Their results demonstrate the effectiveness of higher pressure to eliminate the STEC contamination, however, the heat treatment makes the product undesirable.  The manuscript is written well and can be accepted with following modifications.

Line 58 : “ The surface of the hide of as many as 90% of cattle may be positive 59 for STEC, though single-figure percentages are more usual” Rephrase for clarity.

Line 60 : The proportion of positive samples  falls to almost zero on chilled carcasses during the course of technological processing [13]. Please provide names of technological processing.

Line -162: “ The results of microbiological examination (see Chapter 2.5)”. What chapter you are referring to?

Results : Table -1 : In method section authors have prepared two inocula (log 5 and log 8), however in result section (table -1) they showed the data for log 3 and log 6 which is confusing. Also the log3 and log 6 sample has the similar (7.07 vs 7.29)  TVC before the treatment. Please Include the possible explanation for that.

Results (Tables) :  data can be expressed as below detection limit (BD), instead of the number. The limit of detection can be given in Method section.

Author Response

Response to Reviewer #2

We would like to thank you for your comments and suggestions that have improved the quality of our manuscript.

Comments and Suggestions for Authors

Following is the review of manuscript entitled “The Effect of High Pressure Processing on the Survival of Non-O157 Shiga  Toxin-producing Escherichia coli in Steak Tartare: The Good- or Best-Case  Scenario?”.  The authors have presented data on treatment of steak tartare under high pressure and in combination of heat treatment. Their results demonstrate the effectiveness of higher pressure to eliminate the STEC contamination, however, the heat treatment makes the product undesirable.  The manuscript is written well and can be accepted with following modifications.

Line 58 : “ The surface of the hide of as many as 90% of cattle may be positive 59 for STEC, though single-figure percentages are more usual” Rephrase for clarity.

Line 60 : The proportion of positive samples  falls to almost zero on chilled carcasses during the course of technological processing [13]. Please provide names of technological processing.

Line -162: “ The results of microbiological examination (see Chapter 2.5)”. What chapter you are referring to?

Results : Table -1 : In method section authors have prepared two inocula (log 5 and log 8), however in result section (table -1) they showed the data for log 3 and log 6 which is confusing. Also the log3 and log 6 sample has the similar (7.07 vs 7.29)  TVC before the treatment. Please Include the possible explanation for that.

Results (Tables) :  data can be expressed as below detection limit (BD), instead of the number. The limit of detection can be given in Method section.

We agree. These comments were corrected as suggested by the reviewer.

Round 2

Reviewer 1 Report

The revised paper is in conditions to be published. The authors have performed the revision formerly suggested.  No further revision is asked.